# Potential of Continuous Electric Current on Biometrical, Physiological and Quality Characteristics of Organic Tomato

Madalin-Vasile Gheorghitoaie [1], Ilie Bodale [2], Vladut Achitei [1], Gabriel-Ciprian Teliban [1], Alexandru Cojocaru [1], Gianluca Caruso [3], Gabriela Mihalache [1,4] and Vasile Stoleru [1,*]

[1] Department of Horticulture, "Ion Ionescu de la Brad" Iasi University of Life Sciences, 3 M. Sadoveanu Alley, 700440 Iasi, Romania; madalin.gheorghitoaie@uaiasi.ro (M.-V.G.); achitei.vladut@yahoo.com (V.A.); gabrielteliban@uaiasi.ro (G.-C.T.); acojocaru@uaiasi.ro (A.C.); gabriela.mihalache@uaic.ro (G.M.)

[2] Department of Sciences, "Ion Ionescu de la Brad" Iasi University of Life Sciences, 3 M. Sadoveanu Alley, 700440 Iasi, Romania; ilie.bodale@uaiasi.ro

[3] Department of Agricultural Sciences, University of Naples Federico II, Via Università 100, 80055 Portici, Italy; gcaruso@unina.it

[4] Integrated Center of Environmental Science Studies in the North Eastern Region (CERNESIM), "Alexandru Ioan Cuza" University of Iasi, 11 Carol I, 700506 Iasi, Romania

\* Correspondence: vstoleru@uaiasi.ro

**Abstract:** The tomato is one of the most important species in the food sector. For farmers, the increase in yield in greenhouse conditions by keeping a high quality of fruits represents a goal which is very difficult to achieve in these conditions. Therefore, the present study evaluates the influence that a continuous electric current might have on some biometrical, physiological and quality parameters of tomato fruits. The study was carried out in a greenhouse where tomato plants belonging to Qualitet $F_1$ hybrid were treated with different continuous electric currents, under 5 DC sources, stabilized by the laboratory 0–30 V/0–5 A. During the research, the tomato plants were exposed to different electric current intensities or voltages on the plants or in the soil (T1-0.15 A; T2-0.30 A; T3-0.45 A; T4-1.5 V; T5-1.5 V-soil). The tomato plant samples were compared with an untreated control. In order to determine the influence of a continuous electric current, observations and determinations were made on tomato plants and fruits. The results highlighted significant differences between the treated and not treated plants, regarding the plant height, yield, firmness, acidity, total soluble solids, antioxidant activity, crude and dietary fibres, tannins, oxalates, saponins, $\alpha$-amylase inhibitors, K, Mg, Fe and Zn content. Depending on the intensity of the electric current and the manner of application, the biometrical, physiological and quality parameters of tomato fruits were differently influenced. Both positive and negative influences were registered. More experiments are needed in order to establish a relation between the electric current intensity and the manner of application which can lead to better and higher tomato yields and quality in greenhouse conditions.

**Keywords:** tomato; continuous electric current; quality; biometrical; physiological

## 1. Introduction

Current production policies aim as much as possible to reduce chemical inputs in order to avoid soil, air and crop pollution in sustainable farm conditions [1]. Among the vegetable species worldwide, tomatoes are the most cultivated vegetables with areas exceeding 5.03 million ha and with average yields of 35.9 t·ha$^{-1}$. The expansion of tomato areas is increasing [2]. Worldwide, the average consumption of tomatoes is 21.17 kg/capita [3], and in Romania the average consumption grows to 41.40 kg·capita$^{-1}$ [4].

Nowadays, there is a global interest in obtaining higher tomato production using different methods to improve the absorption of essential minerals in plants in order to increase production efficiently.

Natural or artificial stimulants (physical, chemical) are factors that induce various changes in plant response to their defence by: the accumulation of primary or secondary metabolic compounds (polyphenols, lycopene, proteins, lipids, carbohydrates); physiological changes (chlorophyll compounds, photosynthetic activity, $CO_2$ exchange); and morphological changes (number of fruits, fruit mass, leaf area, production, etc.), in order to optimally restore [5–7].

Electrostimulation of plants can induce plant movement, the activation of ion channels, ion transport, gene expression, enzymatic system activation, electrical signalling, plant-cell damage, enhanced wound healing, and can also influence plant growth [8].

Over the years, various researchers have looked at the ability of plants to generate higher yields through the use of electrical stimuli [6].

Numerous studies that use the effects of electric current on plants starting from the seeds [9], roots [10], cells [11], conservation capacity [12], the uses of chemical nutrients monitored by electric signal [13] and last but not least the production capacity [14] were conducted capturing the attention of researchers.

In the case of the tomato, the positive effects of direct electric current (DC) with an intensity between 15 and 3 µ A on growth and ion accumulation in a tomato plant was studied by Black et al. [15]. In another study, the use of 4–12 kV electricity for 30–40 s led to an increase in seed germination [16]. De Souza et al. by applying a low-voltage electric current 160 mT, 60 Hz, for 1 min on tomato seeds, observed an improvement in the process of germination, a more abundant sprouting, a development of the root system and implicitly an increase in the leaf surface [17].

The purpose of this study was to evaluate the characteristics that tomato fruits have acquired by applying a direct electric current, following the evolution of some biometrical, physiological and quality indicators.

## 2. Materials and Methods

### 2.1. Plant Material and Growth Conditions

The experiment was carried out in the greenhouse at "V. Adamachi" farm from Iasi, under controlled conditions, during 2019–2020. During the research, a Qualitet $F_1$ hybrid with a determined growth was used. The dominant feature of this hybrid is that it lends itself well to growing conditions in protected areas [18].

The sowing took place on 15th of February every year. The seeds were sown in a plastic alveolar tray at 22 °C, 10 h—10,000 Lux, 75% relative humidity (Rh), in a growth chamber. During BBCH 101 phenophases (first true leaf), the seedlings were transplanted into the pots with 540 cm$^3$, using a disinfected substrate of Kekkila$^®$ peat. The substrate used for sowing had the following characteristics: 0–5 mm size; 5.5–5.8 pH; NPK complex 14–16–18+ microelements; wetting agent; EC = 2.5 dS·m$^{-1}$.

At the age of 21 days old, the pots were moved into the greenhouse, under the following conditions: 20–23 °C/16–18 °C day/night temperature; 14 h/10 h light/dark cycle; 76% mean Rh. At 50 days old BBCH 501(first inflorescence visible), the tomato plants were transferred into 12 L plastic pots containing a mix substrate with a pH of 6.10 (85% peat, 10% compost and 5% perlite). During the vegetation period, the substrate was improved five times with organic fertilizer by using 5 kg of Orgevit$^®$ per tons of substrate to optimize the content of macro and microelements for plant growing [1]. The 1st application was at the same time as the planting period, the 2nd time corresponded with the second inflorescence visible (BBCH 502) and the last three applications corresponded with BBCH 503, BBCH 504 and BBCH 505.Tomato plants were given the same amount of water daily, ranging from 1–1.5 L·plant$^{-1}$. During the experiment, growing practices (training, pruning and treatments for pests and diseases) were applied to all the plants [1]. The tomato crop experiment ended on 31 of July.

### 2.2. Design of Experiment

The experiment was carried out using 90 tomato plants divided into 5 treated versions and a control. The experiment was carried out in a split plot design with 3 replications of 5 plants per repetition.

The equipment consists of 5 DC sources stabilized by the laboratory 0–30 V/0–5 A; electrical conductors with a length of 4.5 m, arranged in the form of a spiral with a diameter of 5 cm, these having a resistance of 1.3 Ω, the two ends of the spiral being each connected to a different electric conductor with a length of 4 m and 1.7 Ω resistance; 3 electrical resistors of 20 W and 15 Ω that were used in the case of variants 1, 2 and 3, these being connected at the output of the terminal + within the circuit.

The voltage sources were adjusted in such a way as to provide currents of different intensities in the electrical circuit created, linking in parallel to the terminals of the laboratory source the electrical conductors for each of the variants 1, 2 and 3 (Figure 1).

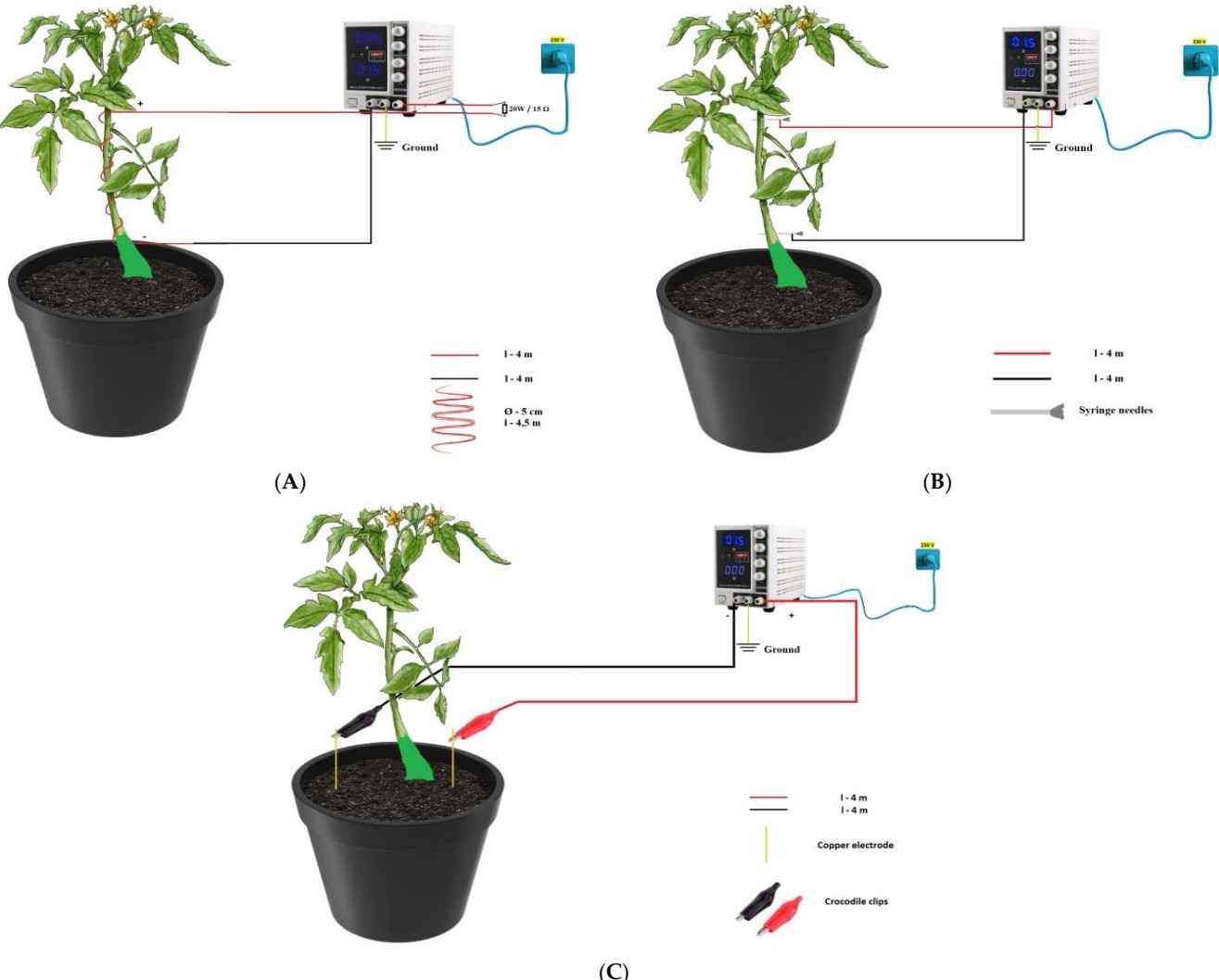

(A)

(B)

(C)

**Figure 1.** The electrical schematic diagram used to provide currents of different intensities to tomato plants ((**A**) = electrical conductor with a length of 4.5 m, arranged in the form of a spiral with a diameter of 5 cm. The DC power applied was 0.15, 0.30, 0.45 A; (**B**) = syringe needles that crossed the plant stem, inserted in two points: in the apical area, and at the base of the stem. The DC power applied was 1.5 V, DC (polarity); (**C**) = two copper electrodes inserted in the soil using crocodile clips (soil version). The DC power applied was 1.5 V.).

A low intensity DC was used in order to prevent plant death since the polarization time up to membrane breakdown is influenced by the field strength [19].

The description of the experimental variants can be found in Table 1.

**Table 1.** Description of experimental variants.

| Treatment | Description of Treatment |
|---|---|
| Treatment 1 (T1) | Applied electric current crossed the circuit, created using a current with an intensity of 0.15 A, using an electrical conductor with a length of 4.5 m, arranged in the form of a spiral with a diameter of 5 cm. |
| Treatment 2 (T2) | Applied electric current crossed the circuit, created using a current with an intensity of 0.30 A, using an electrical conductor with a length of 4.5 m, arranged in the form of a spiral with a diameter of 5 cm. |
| Treatment 3 (T3) | Applied electrical current that crossed the circuit, created using a current with an intensity of 0.45 A, using an electrical conductor with a length of 4.5 m, arranged in the form of a spiral with a diameter of 5 cm. |
| Treatment 4 (T4) | In this case, syringe needles were used that crossed the plant stem, these being inserted twice for each plant, one being inserted in the apical area, and the other at the base of the stem. The electrical conductors were connected to the syringe needles with the help of crocodile clips. The DC power applied was 1.5 V, DC (polarity). |
| Treatment 5 (T5) | In this case, for each plant we inserted two copper electrodes into the soil which were connected to a DC of 1.5 V using crocodile clips (soil version). |
| Control (C) | Control without electric current applied. |

In the case of the 4th treatment (polarity), syringe needles were used that crossed the plant stem, one being mounted in the apical area, and the other at the base of the stem. In this variant, the 4.5 m wire was not used as a spiral, but only the 2 wires with the length of 4 m and the resistance of 1.7 $\Omega$, the electrical conductors being connected to the two syringe needles with the help of crocodile clips. The DC power applied was 1.5 V, DC (Figure 1).

For the 5th treatment (soil variant), the entire circuit was mounted using the same conditions as in the case of the 4th variant, the difference being that the two electrical conductors were connected to 2 electrodes that were inserted into the soil (Figure 1).

Modern cultivation technologies were applied to tomato plants removing tomato suckers [20] and the bottom two leaves [21], and in mid-July the apical area was pruned to stop the vegetative growth on all plants [22].

Fruits were harvested step by step during the experiment, according to full maturity. To control pests and diseases organic measures were applied according to Munteanu [23]. During the vegetation period, 175 L of water was used for each plot.

The climatic conditions from the greenhouse during the experiment are shown in Table 2.

**Table 2.** Tomato growing conditions (2019–2020).

| Months | Temperature (°C) | | Relative Humidity (%) | | Sunlight (kl) | |
|---|---|---|---|---|---|---|
| | **2019** | **2020** | **2019** | **2020** | **2019** | **2020** |
| February | 18.1 | 18.7 | 75.9 | 76.4 | 24.6 | 23.9 |
| March | 20.0 | 18.7 | 73.1 | 71.9 | 39.8 | 35.6 |
| April | 20.3 | 18.9 | 69.3 | 68.4 | 44.6 | 58.4 |
| May | 20.9 | 19.7 | 65.7 | 66.2 | 48.5 | 45.7 |
| June | 26.1 | 24.8 | 65.8 | 71.2 | 60.4 | 53.8 |
| July | 27.5 | 25.6 | 66.2 | 67.2 | 61.9 | 61.7 |

*2.3. Samples Preparation*

From each plot, 10 ripe fruit random samples were collected from the 2nd to 4th clusters for laboratory analyses. The samples needed for biochemical analyses (lycopene, polyphenol and antioxidant capacity) were prepared as follows: fruits were cut into 1 cm fragments and dried on a Sanyo stove, type MOV-112F, at a temperature of 70 °C until constant weight. The samples were then ground into small fragments of 0.1–1 mm [24].

### 2.4. Biometric Measurements

Regarding the tomato production, the parameters that indicate the height of the plants, number of fruits, the average mass of a fruit and the diameter of the fruits were monitored in order to estimate total yield.

The yield (kg·ha$^{-1}$) was calculated using the following formula: (plants/ha × fruits/plant × average fruits weight)/1000 [25]. The plant heights (cm) were measured after the last harvest in each experimental treatment with electric current.

### 2.5. Physiological—Chlorophyll Content

The chlorophyll content was done using the portable chlorophyll content meter (CCM-200 plus Opti-Sciences Chlorophyll Content Meter), for the three leaves specific to the cluster 3, before the fruit is harvested (BBCH 803). The readings were expressed as CCI units.

### 2.6. Organoleptic Properties—Firmness, Total Soluble Solids and Acidity

The firmness or consistence of the tomato fruits was determined using a penetrometer in order to evaluate the degree of maturity or to establish the optimal moment of harvest, according to the harvest destination [26].

The total soluble solids of tomato juice were measured with a digital refractometer (RX5000$\alpha$, Atago, Tokyo, Japan).

The acidity, in terms of content of citric, was quantified by titration using a titrator (808 Titrando, Metrohm, Wesbury, NY, USA) using 6.0 g of tomato juice, which was diluted with 60 mL of ultra clean DI water [27].

### 2.7. Antioxidant Status—Lycopene, β-Carotene, Phenolic Compounds and Antioxidant Activity

The lycopene content was determined according to the spectrophotometric method described by Davis et al. [28]. The β-carotene was determined according to the method described by Cadoni et al. [29]. The content of phenolic compounds was measured using the HPLC-MS method described by Mocan et al. [30], which consists of the detection of different phenolic compounds using a single column. The antioxidant activity was done by following the procedure described by Re et al. [31], and the results obtained were calculated according to Ouis and Hariri [32].

### 2.8. The Proximate Composition—Ash, Crude Lipid, Crude Protein, Crude Fibre, Dietary Fibre, Calorific Value

The proximate composition of tomato fruits was done as follows: the ash content and the calorific value were determined according to AOAC, 2005 [33]; the crude lipid, crude protein, crude fibre and dietary fibre were measured according to AOAC, 2000 [34].

### 2.9. The Antinutritional Composition—Phytate, Tannins, Oxalates, Saponins, Trypsin Inhibitors, α-Amylase Inhibitors

The antinutritional composition of tomato fruits was determined by following different methods: the phytate content was measured according to the method described by Lucas and Markaka [35]; the tannins were determined using the method of Ogawa and Yazaki [36]; the oxalate content was measured using the methods of Chai and Liebman [37] and Umogbai et al. [38], and the results were calculated according to Attalla et al. [39] and Chai and Liebman [40]; the saponins content were measured according to AOAC, 1990 [41]; the trypsin inhibitors were determined as described by Korsinczky et al. [42] and calculated according to Lukanc et al. [43]; the α-amylase inhibitors were measured as described by Barrett and Udani [44].

### 2.10. The Mineral Content

The mineral content of tomato plants was determined by atomic absorption spectrophotometry method Caruso et al. [45].

*2.11. Statistical Analysis*

The data are expressed as the means ± standard deviation (SD). One-way analysis of variance (ANOVA) was used to see the influence of the treatments on the morphological, physiological and quality traits of tomato fruits cv. Qualitet $F_1$. The significant differences between treatments were established using Tukey's post hoc test with a degree of confidence of 95% using a SPSS ver. 21.

## 3. Results and Discussion

To evaluate the influence of continuous electric current on the characteristics of tomato fruits, determinations on the biometrical and physiological features of tomato plants, also on the quality of production were done.

One of the biometrical features of tomato plants that have been carefully monitored refers to the height of the plants (Table 3). This parameter is very important because the more vigorous the plants, the more they have the ability to generate higher yields due to the fact that they will have a larger foliar surface that will be able to convert solar energy into photosynthesis. The results showed that, for the treated plants, the average height was between 154.33 and 161.33 cm, compared to control where the average plant height was only of 125.67 cm. Significant differences were observed between T1 and T2 compared to the control (C). The differences obtained between the treatments of approximately 33% can be attributed to the electric current applications [46]. No differences were registered between the different electric current intensities or the way of its application.

**Table 3.** The biometric measurements of tomato plants and fruits under the influence of different electric currents treatments.

| Variant | Plant Height (cm) | Number of Fruits | Average Fruits Weight (g) | Average Fruits Diameter (mm) | Yield (t·ha$^{-1}$) |
|---|---|---|---|---|---|
| T1 | 161.33 ± 4.48 a | 33.00 ± 2.31 ns | 164.38 ± 11.98 ns | 69.29 ± 1.79 ns | 142.47 ± 35.03 a |
| T2 | 160.33 ± 8.41 a | 36.67 ± 3.76 ns | 148.00 ± 10.65 ns | 68.16 ± 3.31 ns | 138.40 ± 12.89 ab |
| T3 | 154.33 ± 6.96 ab | 38.33 ± 4.37 ns | 132.35 ± 9.52 ns | 64.07 ± 1.70 ns | 133.60 ± 42.76 ab |
| T4 | 156.67 ± 4.91 ab | 36.33 ± 5.21 ns | 147.30 ± 10.59 ns | 67.42 ± 3.49 ns | 138.01 ± 39.45 ab |
| T5 | 155.67 ± 6.12 ab | 36.00 ± 2.52 ns | 124.55 ± 8.96 ns | 63.62 ± 1.18 ns | 115.50 ± 6.32 ab |
| C | 125.67 ± 9.94 b | 27.67 ± 3.48 ns | 118.89 ± 8.55 ns | 62.01 ± 1.91 ns | 83.49 ± 8.30 b |

Variants: T1—0.15 A; T2—0.30 A; T3—0.45 A; T4—polarity; T5—soil version; C—Control; within each column, values associated with different letters are significantly different according to Tukey's test at $p \leq 0.05$; a—high value; ns—nonsignificant.

No significant differences were registered for the number of fruits, the average fruits weight and the average fruits diameter. The results registered for the treated plants were relatively similar to the control, nonsignificant differences being recorded.

Regarding the yield, in the case of all variants where an electric current was applied, higher values were recorded as compared to the control (C). However, significant differences were registered just only between the yield of T1 and the control (C).

The physiological parameter monitored during the experiment was the chlorophyll. Usually, the chlorophyll content of leaves is a key indicator, which is often associated with the yield [47,48]. The chlorophyll content of the plants is shown in Figure 2. Thus, it varied from 32.74 CCI, in the case of T2, to 38.75 CCI, in the case of T3, the differences of approximately 18.47% is not significantly different for $p < 0.05$ as compared to the control (C). Therefore, the application of various intensities of electric currents to the plants did not influence the chlorophyll content of the leaves. Our results suggest that the chlorophyll content did not influence the yield, the lowest yield being recorded in control plants which had a high chlorophyll value.

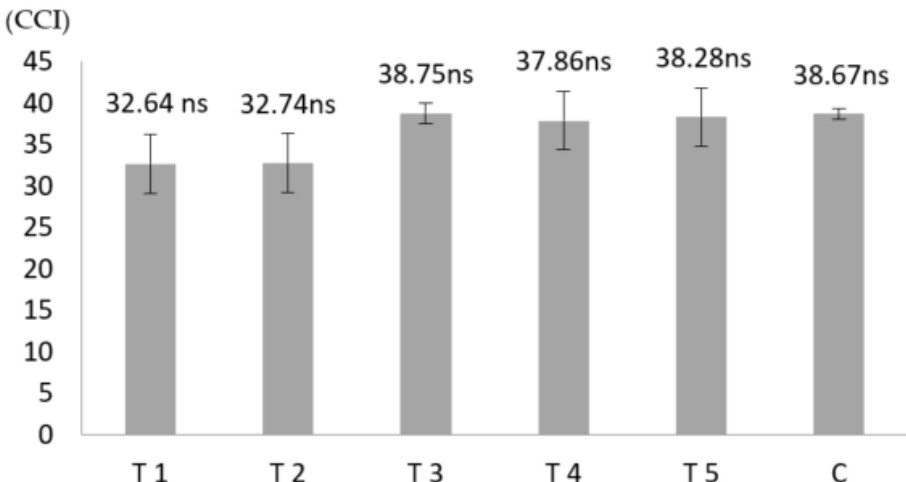

**Figure 2.** Chlorophyll content index from tomato plant (T1—0.15 A; T2—0.30 A; T3—0.45 A; T4—polarity; T5—soil version; C—Control; ns-nonsignificant).

The quality of the tomato fruits was evaluated taking into account some organoleptic properties (firmness, acidity or total soluble solids), the antioxidant status (lycopene, β carotene, polyphenols, antioxidant activity), the proximate composition (crude lipid, protein, fibre, dietary fibre, calorific value), the antinutritional composition (phytate, tannins, oxalate, saponins, trypsin inhibitor, α-amylase inhibitor) and the mineral content (K, Ca, Mg, Fe, Zn, Na).

Firmness is an important quality parameter in terms of storage, consumer acceptance and transportation of tomato fruits. The results of our experiment regarding the firmness of tomato fruits varied from 3.45 kg·cm$^{-2}$ for T1 to 4.93 kg·cm$^{-2}$ for C, significant differences being registered between these treatments. For the rest of the treatments the electric current applied to tomato plants did not significantly influence the firmness of the tomato fruits as compared to the control. A reduction in firmness is usually associated with the dissociation of middle lamella and production of a variety of enzymes which affect the cell wall integrity [49,50]. In our experiment it was observed that the application of the lowest electric current (T1—0.15 A) decreased the most the firmness of tomato fruits. However, the differences between T1 and the rest of the treatments with electric current were not significant. In comparison with other studies which showed a decrease in the firmness along with an increase in the electric treatment intensity [51,52], the application of 0.30 A (T2); 0.45 A (T3) or 1.5 V (T4 and T5) did not affect the firmness of tomato fruits.

Regarding the influence of the treatments applied to the plants on the acidity of tomato fruits, results ranged from 0.4 g citric acid·100 g$^{-1}$ f.w. (T5), to 0.55 g citric acid·100 g$^{-1}$ f.w. (T3) (Table 4). A significant higher acidity of the fruits as compared with the control (0.44 g citric acid·100 g$^{-1}$ f.w.) was registered for T3 (0.55 g citric acid·100 g$^{-1}$ f.w.), T1 and T2 (both of 0.53 g citric acid·100 g$^{-1}$ f.w.), while for T4 and T5 the acidity was almost the same as in the case of the control. According to Anthon et al., [53], the acidity of tomato fruits can be used as an indicator of maturity, as the content of acids decrease along with the ripening of the fruits. During the ripening of tomato fruits, citric acid is the one that last disappear [54]. Therefore, the lower the value of citric acid, ripen the tomato. This suggest that in our experiment, at the time of harvest, the tomatoes fruits of T4 and T5 were better ripen than the rest of the fruits belonging to the plants subjected to the electric current treatments.

**Table 4.** The organoleptic properties of tomato fruits under the influence of different electric current treatments.

| Variant | Firmness (kg·cm$^{-2}$) | Acidity (g Citric Acid·100 g$^{-1}$ f.w.) | Total Soluble Solids (° Brix) |
|---|---|---|---|
| T1 | 3.45 ± 0.25 b | 0.53 ± 0.03 a | 5.23 ± 0.03 a |
| T2 | 3.82 ± 0.28 ab | 0.53 ± 0.01 a | 5.07 ± 0.03 ab |
| T3 | 4.02 ± 0.29 ab | 0.55 ± 0.01 a | 5.03 ± 0.03 b |
| T4 | 4.56 ± 0.33 ab | 0.43 ± 0.02 b | 4.83 ± 0.03 c |
| T5 | 3.87 ± 0.28 ab | 0.40 ± 0.01 b | 5.07 ± 0.03 ab |
| C | 4.93 ± 0.35 a | 0.44 ± 0.02 b | 5.10 ± 0.06 ab |

Different letters (a–c) in a column are significantly different from each other ($p \leq 0.05$) obtained from Tukey's multiple comparison test; a—high value; T1—0.15 A; T2—0.30 A; T3—0.45 A; T4—polarity; T5—soil version; C—Control.

The results of total soluble solids showed that the electric current applied to tomato plants did not influence the amount of total soluble solids, the registered values being not significantly different as compared with the control, except the plants subjected to 1.5 V where one syringe needle was inserted in the apical area and another at the base of the stem (T4) for which a significant decrease in the total soluble solids was recorded (Table 4). The lack of differences between the treated plant and the control can suggest that the electric current intensities used in the experiment did not affect the cell activity or metabolism. In other experiments where electric current pulses were applied to tomato plants, the increase in the total soluble solids content was linked to various causes: (1) osmotic imbalance, which led to the accumulation of sugars in order to restore the cell activity, (2) acceleration of ripening as a result of increased metabolic activity, (3) disarrangements of the polysaccharides and molecular bonds found in the cell wall [52].

The effect of continuous electric current treatment on the content of lycopene, polyphenols and the antioxidant capacity of tomato fruits is presented in Table 5.

**Table 5.** The antioxidant status of tomato fruits.

| Variant | Lycopene (mg·100 g$^{-1}$ d.w.) | β-Carotene (mg·100 g$^{-1}$ d.w.) | Polyphenols (mg·100 g$^{-1}$ d.w.) | Antioxidant Activity (mmol Trol·100 g$^{-1}$ d.w.) |
|---|---|---|---|---|
| T1 | 8.21 ± 0.68 ns | 7.46 ± 0.49 ns | 1698.45 ± 83.50 ns | 83.05 ± 0.51 a |
| T2 | 9.16 ± 0.81 ns | 7.69 ± 0.50 ns | 1722.42 ± 199.81 ns | 77.03 ± 0.56 b |
| T3 | 10.52 ± 0.82 ns | 7.47 ± 0.49 ns | 1863.18 ± 226.02 ns | 72.84 ± 0.15 c |
| T4 | 9.78 ± 1.32 ns | 7.58 ± 0.50 ns | 1766.87 ± 100.22 ns | 73.28 ± 0.34 c |
| T5 | 9.43 ± 0.53 ns | 7.77 ± 0.51 ns | 1772.43 ± 104.44 ns | 73.91 ± 0.68 bc |
| C | 7.85 ± 0.66 ns | 7.00 ± 0.46 ns | 1735.11 ± 33.10 ns | 81.35 ± 1.35 a |

Different letters (a–c) in a column are significantly different from each other ($p \leq 0.05$) obtained from Tukey's multiple comparison test; a—high value; ns—nonsignificant; T1—0.15 A; T2—0.30 A; T3—0.45 A; T4—polarity; T5—soil version; C—Control.

The lycopene content varied from 7.85 mg·100 g$^{-1}$ d.w., in the control to 10.52 mg·100 g$^{-1}$ d.w., under treatment of DC with 0.45 A (T3); the difference of 34% being not significant for $p < 0.05$. However, the treatments with electric current had a positive influence on the lycopene content of tomato fruits, with an increase in its content being observed alongside the increase in the intensity of electric current (T1 < T2 < T3). T4 and T5, which consisted of an electric power of 1.5 V, gave approximately the same lycopene content (9.78 mg·100 g$^{-1}$ d.w. and 9.43 mg·100 g$^{-1}$ d.w., respectively). An increase in the lycopene content due to pulsed electric fields was also recorded by Vallverdu-Queralt et al. [55]. The mechanism proposed was the ethylene production as a result of the stress determined by pulsed electric fields which activated the enzymes involved in lycopene biosynthesis [53]. Thus, the increase in lycopene content in our experiment might be the result of an increased ethylene production in the plants subjected to an electric current which enhanced the lycopene content.

The β carotene content of the treated plants, as in the case of lycopene content, was not significant different as compared to the control. The values recorded varied from 7 mg·100 g$^{-1}$ d.w. to 7.77 mg·100 g$^{-1}$ d.w. Even if the differences between treatments were not significant, the plants subjected to electric current had tomato fruits with higher β carotene values as compared to the control. The treatment in which two copper electrodes connected to an electric current intensity of 1.5 V were inserted in the soil (T5) had the best positive influence on the β carotene content. The increase in the β carotene content due to the application of an electric current might be the result of an increased influx of $Ca^{2+}$ through cations channels, due to an increase at the cellular level of the membrane permeability for $Ca^{2+}$, which promotes the biosynthesis of carotenoids in tomato fruits [55].

Regarding the influence of the treatments on the polyphenol content, the results were non-significant for $p < 0.05$. The registered values ranged from 1698.45 mg 100 g$^{-1}$ d.w., in T1, to 1863.18 mg 100 g$^{-1}$ d.w., in T3. Similar results in tomatoes were obtained by Vallverdu-Queralt et al. [55], in tuberous roots of radish by Dannehl et al. [56], in lettuce by Yu et al. [57] and watercress by Dannehl et al. [58]. As in the case of lycopene and β carotene contents, the content of polyphenols in tomato fruits increased as the intensity of the electric current was risen, the influence of the treatments with 1.5 V (T4 and T5) being almost the same as in the case of 0.30 A (T2).

The continuous electric current treatments applied to the plants registered significant differences on the antioxidant capacity (Tabel 5). This ranged from 72.84 mmol Trol 100 g$^{-1}$ d.w., in the case of T3, to 83.05 mmol Trol · 100 g$^{-1}$ d.w., in the case of T1. No significant differences were registered between T1 and C, while for the rest of the treatments with electric current, the antioxidant activity was significantly lower than in the case of the non-treated plants (C). The results obtained in our experiments were lower, than those recorded by Vallverdu-Queralt et al.: 101.60 mmol Troll 100 g$^{-1}$ f.w., in the case of treatment with 2 kV·cm$^{-1}$ for 5 pulses, up to 134.67 mmol Troll 100 g$^{-1}$ f.w., in the case of treatment with 1.2 kV·cm$^{-1}$ for 30 pulses [55].

The proximate composition of tomato fruits is presented in Table 6. The electric current treatments did not influence the content of the ash, crude lipid, crude protein and the calorific value, no significant differences being registered as compared with the control. Only for the crude fibre and dietary fibre significant differences were obtained between the treatments. The crude fibre content varied from 13.43 g·100 g$^{-1}$ d.w. for C and T2 to 23.07 g·100 g$^{-1}$ d.w. for T3. Except for the treatments T3 and T4 for which significant differences were registered as compared with the C, no significant differences were recorded for the rest of the treatments. Regarding the dietary fibre content of tomato fruits, the plants subjected to 0.15 A (T1) and 1.5 V—soil version (T5) registered significantly higher values as compared with the control. No significant differences were registered between the rest of the electric current treatments and the control. The dietary fibres are mainly composed of cellulose, hemicelluloses, lignin, pectins, resistant starch and non-digestible oligosaccharides. Cellulose which accounts for about 33% of the cell wall of fresh fruit can fluctuate in quantity during the ripening of the fruits. The consumption of foods rich in dietary fibres is very important taking into account their function in the normal peristaltic movement of the bowels [59]. Therefore, in our experiment, the increase in the amount of dietary fibre in the tomato fruits of the plants subjected to different intensities of electric current represents a positive feature for consumption.

The antinutritional composition of tomato fruits subjected to different electric current treatments is presented in Table 7. The results show that the content of phytate and trypsin inhibitor are not influenced by the electric current treatments, regardless of the intensity applied. Influences of the electric current application were registered for the content of tannins, oxalate, saponin and α-amylase inhibitor.

**Table 6.** The proximate composition of tomato fruits under different electric current treatments.

| Variant | Ash (g·100 g$^{-1}$ d.w.) | Crude Lipid (g·100 g$^{-1}$ d.w.) | Crude Protein (g·100 g$^{-1}$ d.w.) | Crude Fibre (g·100 g$^{-1}$ d.w.) | Dietary Fibre (g·100 g$^{-1}$ d.w.) | Calorific Value kJ/g (kcal/100 g) |
|---|---|---|---|---|---|---|
| T1 | 4.51 ± 0.29 ns | 11.29 ± 0.74 ns | 33.44 ± 2.19 ns | 15.43 ± 1.01 bc | 9.52 ± 0.62 a | 441.95 ± 28.97 ns |
| T2 | 4.11 ± 0.27 ns | 11.41 ± 0.75 ns | 31.96 ± 2.09 ns | 13.43 ± 0.88 c | 6.74 ± 0.44 b | 436.42 ± 28.60 ns |
| T3 | 4.83 ± 0.32 ns | 11.90 ± 0.78 ns | 31.27 ± 2.05 ns | 23.07 ± 1.51 a | 8.30 ± 0.54 ab | 423.91 ± 27.78 ns |
| T4 | 4.34 ± 0.28 ns | 12.25 ± 0.80 ns | 29.49 ± 1.93 ns | 20.02 ± 1.31 ab | 8.71 ± 0.57 ab | 423.98 ± 27.78 ns |
| T5 | 4.76 ± 0.31 ns | 11.70 ± 0.77 ns | 32.54 ± 2.13 ns | 15.83 ± 1.04 bc | 9.55 ± 0.63 a | 452.68 ± 29.67 ns |
| C | 5.12 ± 0.34 ns | 11.21 ± 0.74 ns | 30.96 ± 2.03 ns | 13.43 ± 0.88 c | 6.74 ± 0.44 b | 445.75 ± 29.21 ns |

Different letters (a–c) in a column are significantly different from each other ($p \leq 0.05$) obtained from Tukey's multiple comparison test; a-high value; ns—nonsignificant; T1—0.15 A; T2—0.30 A; T3—0.45 A; T4—polarity; T5—soil version; C—Control.

**Table 7.** The antinutritional composition of tomato fruits under different electric current treatments.

| Variant | Phytate (g·100 g$^{-1}$ d.w.) | Tannins (g·100 g$^{-1}$ d.w.) | Oxalates (g·100 g$^{-1}$ d.w.) | Saponins (g·100 g$^{-1}$ d.w.) | Trypsin Inhibitors (TUI mg$^{-1}$) | α-Amylase Inhibitors IC$_{50}$ (mg ml$^{-1}$) |
|---|---|---|---|---|---|---|
| T1 | 3.75 ± 0.25 ns | 3.96 ± 0.26 a | 2.14 ± 0.14 bc | 2.63 ± 0.17 bc | 9.26 ± 0.61 ns | 0.77 ± 0.05 a |
| T2 | 3.59 ± 0.23 ns | 2.46 ± 0.16 cd | 1.36 ± 0.09 c | 1.66 ± 0.11 d | 9.95 ± 0.65 ns | 0.43 ± 0.03 bc |
| T3 | 3.43 ± 0.23 ns | 1.77 ± 0.12 de | 3.84 ± 0.25 a | 3.94 ± 0.26 a | 9.75 ± 0.64 ns | 0.35 ± 0.02 c |
| T4 | 3.27 ± 0.22 ns | 1.18 ± 0.08 e | 4.00 ± 0.26 a | 3.24 ± 0.21 ab | 9.42 ± 0.62 ns | 0.54 ± 0.03 b |
| T5 | 4.02 ± 0.27 ns | 3.46 ± 0.23 ab | 2.48 ± 0.16 b | 2.82 ± 0.18 b | 9.49 ± 0.62 ns | 0.82 ± 0.05 a |
| C | 3.89 ± 0.25 ns | 2.65 ± 0.17 bc | 1.57 ± 0.10 c | 1.78 ± 0.12 cd | 10.07 ± 0.66 ns | 0.53 ± 0.03 bc |

Different letters (a–e) in a column are significantly different from each other ($p \leq 0.05$) obtained from Tukey's multiple comparison test; a—high value; ns—nonsignificant; T1—0.15 A; T2—0.30 A; T3—0.45 A; T4—polarity; T5—soil version; C—Control.

The content of tannins varied between 1.18 g·100 g$^{-1}$ d.w. and 3.96 g·100 g$^{-1}$ d.w. According to the results, the application of an electric current intensity of 0.45 A (T3) or 1.5 V—the polarity version (T4) lowered the content of tannins in tomato fruits; the values registered being significant lower as compared to the control (2.65 g·100 g$^{-1}$ d.w.). The decrease in the amount of tannin in tomato fruits is a positive fact taking into account the negative effects that they can have on health [60]. No significant differences were registered between the control and T2 (0.30 A) or T5 (1.5 V—soil version). The highest content of tannins was registered at the lowest intensity of the electric current (0.15 A)—3.96 g·100 g$^{-1}$ d.w. (T1).

The oxalate content in the tomato fruits of the plants treated with electric current of 0.15 A (T1) and 0.30 A (T2) intensity was not significant different as compared with the value recorded for the tomato fruits of the control plants. However, at 1.5 V intensity of electric current (T3—0.45 A; T4—polarity; T5—soil version), the values registered were significant higher. The consumption of high amounts of oxalate can negatively impact the health by causing nutritional deficiency because of the limited absorption of Ca$^{2+}$, Fe$^{2+}$ and Mg$^{2+}$ ions [13,60].

The saponin content of the tomato fruits ranged from 1.78 g·100 g$^{-1}$ d.w. to 3.94 g·100 g$^{-1}$ d.w. Significant higher contents of saponin as compared to the control were registered for T3, T4 and T5. For the rest of the treatments no significant differences were registered. As in the case of oxalate, the application of 0.45 A and 1.5 V intensities led to the accumulation of higher amounts of saponin. Recently, studies have shown that the presence of oxalate in foods is considered to be beneficial for health due to some properties related to cholesterol reduction, immune system stimulation, peptic ulcer or cancer prevention [60]. Therefore, in our experiment, the increase in the oxalate content in tomato fruits due to the electric current application is a positive aspect.

Regarding the α-amylase inhibitor, the values varied from 0.35 mg ml$^{-1}$ to 0.82 mg ml$^{-1}$. The application of electric current to the tomato plants influenced the content of α-amylase inhibitor from tomato fruits, significant differences being regis-

tered between T5 or T1 (0.82 mg mL$^{-1}$ and 0.77 mg mL$^{-1}$, respectively) and the control (0.53 mg mL$^{-1}$). No significant differences were registered for the rest of the treatments. Usually, the presence of $\alpha$-amylase inhibitor in food has negative impacts on human health causing various digestive disorders [60].

The influence of the electric current application on the mineral content of tomato fruits is presented in Table 8. Additionally, in Table 8 is presented the dry matter and the moisture of the tomato fruits. For both the dry matter and the moisture, no significant differences were registered between the treated plants and the control. The dry matter content of the tomato fruits varied from 5.79%, in the case of T4, to 6.17%, in the case of T3. The results are in accordance with the literature (4.0%, in the case of the 988 tomato lines, up to 9.2%, in the case of Poly20) [61]. The moisture content of the tomato fruits, ranged from 93.83%, in the case of T3, to 94.21%, in the case of T4. These results are between the normal ranges found in commercial varieties of tomato [62].

**Table 8.** The mineral content, dry matter and moisture of tomato fruits.

| Variant | Dry Matter (%) | Moisture (%) | K (ppm) | Ca (ppm) | Mg (ppm) | Fe (ppm) | Zn (ppm) | Na (ppm) |
|---|---|---|---|---|---|---|---|---|
| T1 | 5.82 ± 0.14 ns | 94.18 ± 0.14 ns | 2.02 ± 0.13 d | 61.33 ± 4.02 ns | 30.08 ± 1.97 c | 1.89 ± 0.12 b | 0.59 ± 0.04 bc | 1.33 ± 0.09 ns |
| T2 | 5.95 ± 0.15 ns | 94.05 ± 0.15 ns | 2.73 ± 0.18 cd | 65.56 ± 4.30 ns | 33.47 ± 2.19 bc | 2.35 ± 0.16 b | 0.33 ± 0.02 d | 1.35 ± 0.09 ns |
| T3 | 6.17 ± 0.13 ns | 93.83 ± 0.13 ns | 3.17 ± 0.21 bc | 69.88 ± 4.58 ns | 34.50 ± 2.26 bc | 2.59 ± 0.17 b | 0.53 ± 0.03 bc | 1.37 ± 0.09 ns |
| T4 | 5.79 ± 0.14 ns | 94.21 ± 0.14 ns | 5.55 ± 0.36 a | 75.58 ± 4.95 ns | 45.82 ± 3.00 ab | 4.89 ± 0.32 a | 0.89 ± 0.06 a | 1.52 ± 0.10 ns |
| T5 | 5.89 ± 0.14 ns | 94.11 ± 0.14 ns | 3.96 ± 0.26 b | 71.86 ± 4.71 ns | 49.54 ± 3.25 a | 4.39 ± 0.29 a | 0.67 ± 0.04 b | 1.34 ± 0.09 ns |
| C | 5.91 ± 0.14 ns | 94.09 ± 0.14 ns | 1.72 ± 0.11 d | 76.13 ± 4.99 ns | 43.55 ± 2.85 ab | 2.63 ± 0.17 b | 0.42 ± 0.03 cd | 1.11 ± 0.07 ns |

Different letters (a–d) in a column are significantly different from each other ($p \leq 0.05$) obtained from Tukey's multiple comparison test; a—high value; ns—nonsignificant; T1—0.15 A; T2—0.30 A; T3—0.45 A; T4—polarity; T5—soil version; C—Control.

Regarding the mineral content, the results show that significant differences between the fruits of the treated plants and the control were registered for K, Mg, Fe and Zn. The content of Ca and Na were not influenced by the electric current.

The highest content of K, Fe and Zn in tomato fruits was registered when the electric current was applied at an intensity of 1.5 V, for both the polarity (T4) and the soil version (T5). Regarding the Mg content, the highest values were registered when the plants were subjected to 0.15 A, 0.30 A and 0.45 A (T1, T2 and T3, respectively). The mineral content increase in the tomato fruits due to the electric current application can be considered normal taking into account that the mineral absorption by plants is an electrical process and the ions are electrically charged [63]. However, according to our results, the uptake and accumulation of different types of minerals were dependent on the manner the electric current was applied. Therefore, a better accumulation of K, Fe and Zn was recorded when the electric current was applied through syringe needles inserted in the apical area and the base of the stem or when both the electrodes were inserted in the soil. On the other hand, magnesium had better uptake and accumulation when the applied electric current crossed the circuit. Regarding the different intensities, there were slight increases in the mineral content, regardless of the elements, as the current intensity increased.

## 4. Conclusions

The application of a continuous electric current at different intensities and in different manners influenced the characteristics of tomato fruits such as the morphological features (plants height), the yield and the quality of tomato fruits (firmness, acidity, total soluble solids, antioxidant activity, crude fibre, dietary fibres, tannins, oxalate, saponin, $\alpha$-amylase inhibitor and the content K, Mg, Fe and Zn). The lowest electric current intensity (T1, 0.15 A) had a significant positive effect on the plant height, yield, antioxidant activity and dietary fibre, but also led to the accumulation of higher amounts of tannins and $\alpha$-amylase inhibitor and increased the acidity of tomato fruits. Regarding the electric current intensity of 0.30 A (T2), a positive influence was observed only on the plant height. The highest electric current intensity used in the experiment (0.45 A, T3) positively influenced the crude fibre, the tannins, the oxalate and the K content, but at the same time, increased the acidity,

the saponin content and decreased the antioxidant activity. The application of 1.5 V, the polarity version (T4), positively influenced the crude fibre, and the content of tannins, oxalate, K, Mg, Fe and Zn. The same electric current version decreased the total soluble, the antioxidant activity and the saponin content. When 1.5 V was applied through soil (T5) a positive influence was recorded for the content of dietary fibres, oxalate, K, Fe and Zn. Additionally, an increase in the saponin or α-amylase inhibitor content and a decrease in the antioxidant activity was observed. Therefore, different electric current intensities and ways of application can have both a positive effect on some characteristics of tomato fruit and a negative one on other traits. More experiments are needed in order to establish a relationship between the electric current intensity and the manner of its application which results in a better tomato fruit quality.

**Author Contributions:** V.S., I.B. and M.-V.G. conceived and planned the experimental protocol; V.S. performed the research supervision; M.-V.G., V.A., G.-C.T. and A.C. carried out the field experiment and determinations; M.-V.G., V.A. and G.M. were involved in laboratory analyses; V.S., G.-C.T. and G.C. contributed to data statistical processing and interpretation; A.C., G.M. and G.-C.T. were involved in the bibliographic search; V.S., M.-V.G., G.M. and G.C. wrote the draft and final manuscript. All authors have read and agreed to the published version of the manuscript.

**Funding:** This research received no external funding.

**Institutional Review Board Statement:** Not applicable.

**Informed Consent Statement:** Not applicable.

**Data Availability Statement:** Not applicable.

**Acknowledgments:** The authors wish to thank "Ion Ionescu de la Brad" Iasi University of Life Sciences for the financial support of the experiments and Monica Butnariu for the supervision analyses.

**Conflicts of Interest:** The authors declare no conflict of interest.

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
