# Peer review of "Potential of Continuous Electric Current on Biometrical, Physiological and Quality Characteristics of Organic Tomato"

_applsci, doi:10.3390/app12094211_

Round 1

Reviewer 1 Report

This manuscript investigated the effect of continuous electric current at different intensities and in different manners on the characteristics of tomato fruits such as the morphological features (plants height), the yield, and the quality of tomato fruits (firmness, acidity, total soluble solids, antioxidant activity, crude fibre, dietary fibres, tannins, oxalate, saponin, α-amylase inhibitor and the content K, Mg, Fe and Zn). The workload was very full and lots of experimental results were also obtained. However, there were short of necessary analysis for the obtained mass results, so this manuscript was more like a lab report, rather than a research paper.

Author Response

Dear Editor,

We thank you for your interest in our paper. We also thank the reviewers for the patient and careful examination of our manuscript and for providing corrections that will improve this manuscript. Our point-by-point responses regarding comments are detailed on the following pages. The suggestion made by Reviewer 3 was highlighted with green. The changes proposed by Reviewer 4 were made with red. Other changes, including the language check were made with orange. We are grateful for the opportunity to resubmit the manuscript.

Answer to Reviewer 1

Comment 1: This manuscript investigated the effect of continuous electric current at different intensities and in different manners on the characteristics of tomato fruits such as the morphological features (plants height), the yield, and the quality of tomato fruits (firmness, acidity, total soluble solids, antioxidant activity, crude fibre, dietary fibres, tannins, oxalate, saponin, α-amylase inhibitor and the content K, Mg, Fe and Zn). The workload was very full and lots of experimental results were also obtained. However, there were short of necessary analysis for the obtained mass results, so this manuscript was more like a lab report, rather than a research paper.

Answer to Comment 1: Thank you for your comment. In the our future experiments we will be more carefull on choosing the analysis.

Reviewer 2 Report

I congratulate the authors for the relevance of the experiment organized, the complexity of the activities carried out and the results obtained.

I suggest to the authors to continue their research on tomatoes as well as other vegetables of interest grown in protected areas (cucumbers, eggplants).

Author Response

Dear Editor,

We thank you for your interest in our paper. We also thank the reviewers for the patient and careful examination of our manuscript and for providing corrections that will improve this manuscript. Our point-by-point responses regarding comments are detailed on the following pages. The suggestion made by Reviewer 3 was highlighted with green. The changes proposed by Reviewer 4 were made with red. Other changes, including the language check were made with orange. We are grateful for the opportunity to resubmit the manuscript.

Answer to Reviewer 2

Comment 1: I congratulate the authors for the relevance of the experiment organized, the complexity of the activities carried out and the results obtained.

I suggest to the authors to continue their research on tomatoes as well as other vegetables of interest grown in protected areas (cucumbers, eggplants).

Answer to Comment 1: Thank you for your remarks. We will take into consideration your suggestion regarding the research on other vegetables of interest grown in protected areas.

Reviewer 3 Report

The article is written correctly. The abstract and keywords are appropriate and the introduction relates to the research problem. The research methods used are appropriate, the aim of the research is correct. The authors presented the influence of the constant electric field on selected properties of tomatoes. Despite the detailed methodological description of the experiment, it is still proposed to include a schematic diagram of the installation for the interaction of a constant electric field and the location of plants in this field. However, it should be concluded that the results are correct and contribute to the development of science and have significant practical importance.

Author Response

Dear Editor,

We thank you for your interest in our paper. We also thank the reviewers for the patient and careful examination of our manuscript and for providing corrections that will improve this manuscript. Our point-by-point responses regarding comments are detailed on the following pages. The suggestion made by Reviewer 3 was highlighted with green. The changes proposed by Reviewer 4 were made with red. Other changes, including the language check were made with orange. We are grateful for the opportunity to resubmit the manuscript.

Answer to Reviewer 3 (green)

Comment 1: The article is written correctly. The abstract and keywords are appropriate and the introduction relates to the research problem. The research methods used are appropriate, the aim of the research is correct. The authors presented the influence of the constant electric field on selected properties of tomatoes. Despite the detailed methodological description of the experiment, it is still proposed to include a schematic diagram of the installation for the interaction of a constant electric field and the location of plants in this field. However, it should be concluded that the results are correct and contribute to the development of science and have significant practical importance.

Answer to Comment 1: Thank you for your remarks. A schematic diagram of the installation was added in the section 2.2. Design of experiment.

Reviewer 4 Report

The authors of this manuscript present an interesting research study regarding the potential of continuous electric current on biometrical, physiological and quality characteristics of organic tomato. Introduction and the main text are well described, however, in my opinion are too short. Presented tables describes the findings of this work. Due to limited research in the specific area, I believe that discussion section is fair and the authors discuss and explain the findings of their work. The text needs few revisions. Research study on the potential of continuous electric current on biometrical, physiological and quality characteristics of organic tomato is not studied well, thus, this research can add further interest for research scientists.

Abstract

COMMENT:

The Abstract section describes sufficiently the findings of this study.

Line 22     “Therefore, the present study follows the influence that the continuous electric current might have on the yield 22 but also on some biometrical, physiological and quality parameters of tomato fruits”….please check this phrase.

Introduction

The Introduction section is well written. However, according to my opinion is too short.

Lines 55-63…..I believe that these paragraphs could not to be separated.

Materials and Methods

Line 112     replace word “To fight pests” with the word “control”

According to my opinion it could be more helpful if you add a picture or a schematic diagram on how you connect your plants with the continuous electric current.

Furthermore, i understand that in this research many parameters are evaluated. However, according to the number of tomato plants mentioned in line 87 I think (it is my opinion) that 90 plants is a small number. I suggest to repeat your research in the future. On the other hand, this research work is “strong” since the authors collect many and different type of measurements from these plants.

Results and Discussion

Line 219     “Anyway” is not the most appropriate word. Please rephrase

Line 259    d.w..

Conclusions

I think that conclusions describe sufficiently the findings of the research work

Please check and complete: Institutional Review Board Statement, Informed Consent Statement and Data Availability Statement

References

Please check again reference style according to the author’s instructions. For i.e

Line 438  Space life sciences: life support systems and biological systems under influence of physical factors      

Line 449   Can. J. Bot.

Line 448   Fensom, D.S.; (do not leave gap)

Line 478    Crantz from Romania and Their Antioxidant and Antimicrobial Properties      (first letter is capitilised)

Line 491, 509, 527    The same as above

Line 505     Macrolepiota procera, Armillaria mellea and Amanita phalloides    (latin names in Italics)

Line 550    Brassica oleracea var. acephala

Author Response

Dear Editor,

We thank you for your interest in our paper. We also thank the reviewers for the patient and careful examination of our manuscript and for providing corrections that will improve this manuscript. Our point-by-point responses regarding comments are detailed on the following pages. The suggestion made by Reviewer 3 was highlighted with green. The changes proposed by Reviewer 4 were made with red. Other changes, including the language check were made with orange. We are grateful for the opportunity to resubmit the manuscript.

Answer to Reviewer 4 (red)

Comment 1: The Abstract section describes sufficiently the findings of this study.

Line 22     “Therefore, the present study follows the influence that the continuous electric current might have on the yield 22 but also on some biometrical, physiological and quality parameters of tomato fruits”….please check this phrase.

Answer to Comment 1: The statement was re-phrased as follows: “Therefore, the present study evaluates the influence that continuous electric current might have on some biometrical, physiological and quality parameters of tomato fruits.” (Line 22)

Comment 2: The Introduction section is well written. However, according to my opinion is too short.

Lines 55-63…..I believe that these paragraphs could not to be separated.

Answer to Comment 2:Add

Answer to Comment 2: An extra paragraph was added to the introduction.

The paragraphs were linked.

Comment 3: Line 112     replace word “To fight pests” with the word “control”

According to my opinion it could be more helpful if you add a picture or a schematic diagram on how you connect your plants with the continuous electric current.

Furthermore, i understand that in this research many parameters are evaluated. However, according to the number of tomato plants mentioned in line 87 I think (it is my opinion) that 90 plants is a small number. I suggest to repeat your research in the future. On the other hand, this research work is “strong” since the authors collect many and different type of measurements from these plants.

Answer to Comment 3: As suggested “To fight pests” was replaced with “control”

A schematic diagram on how the plants were connected to continuous electric current was added in the section 2.2. Design of experiment.

In our future experiments we will try to increase the number of plants used.

Comment 4:  Line 219     “Anyway” is not the most appropriate word. Please rephrase

Line 259    d.w..

Answer to Comment 4: As suggested “Anyway” was replaced with “However”

”f.w.” was replaced with ”d.w.” where was not correct.

Comment 5:  I think that conclusions describe sufficiently the findings of the research work

 Please check and complete: Institutional Review Board Statement, Informed Consent Statement and Data Availability Statement

Answer to Comment 5: We checked the statements and we decided to exclude them since none of them were applicable.

Comment 6: 

Please check again reference style according to the author’s instructions. For i.e

Line 438  Space life sciences: life support systems and biological systems under influence of physical factors     

Line 449   Can. J. Bot.

Line 448   Fensom, D.S.; (do not leave gap)

Line 478    Crantz from Romania and Their Antioxidant and Antimicrobial Properties      (first letter is capitilised)

Line 491, 509, 527    The same as above

Line 505     Macrolepiota procera, Armillaria mellea and Amanita phalloides    (latin names in Italics)

Line 550    Brassica oleracea var. acephala

Answer to Comment 5:

Line 438  Space life sciences: life support systems and biological systems under influence of physical factors      - was replaced with Adv Space Res.

Line 449   Can. J. Bot.  - was made Italic

Line 448   Fensom, D.S.; (do not leave gap) - The gap was deleted, also other gaps were deleted.

Line 478    Crantz from Romania and Their Antioxidant and Antimicrobial Properties      (first letter is capitilised) - The capitals were replaced with small letters.

Line 491, 509, 527    The same as above  - The capitals were replaced with small letters.

Line 505     Macrolepiota procera, Armillaria mellea and Amanita phalloides    (latin names in Italics) - Done

Line 550    Brassica oleracea var. acephala – was made Italic

Round 2

Reviewer 1 Report

The authors have revised the original manuscript very carefully.